# Variational Causal Autoencoder for Interventional and Counterfactual Queries

## Abstract

We propose the *Variational Causal Autoencoder* (VCAUSE), a novel class of variational graph autoencoders for causal inference in the absence of hidden confounders, when only observational data and the causal graph are available. Without making any structural assumption, VCAUSE mimics the necessary properties of a *Structural Causal Model* (SCM) to provide a framework for performing interventions (*do-operator*) and *abduction-action-prediction* steps. As a result, and as shown by our empirical results, VCAUSE provides a practical and accurate pipeline for estimating the interventional and counterfactual distributions of diverse SCMs. Finally, we apply VCAUSE to evaluate counterfactual fairness in classification problems and also to learn accurate and fair classifiers.

## 1 Introduction

Predicting causal effects of actions (interventions) is a central problem in scientific research in a broad variety of fields [4, 5, 7, 23, 51], and machine learning is no exception [44]. As an example, fundamental machine learning questions—such as fairness [6, 9, 19, 24, 25] and interpretability [17]—, are increasingly being formulated as causal queries.

Research on causal reasoning has predominantly focused on *causal discovery*, a.k.a. structure learning, aimed at discovering the underlying causal graph from data (see, e.g., [15, 30, 49, 60]). An alternative line of work instead aims to answer *causal queries* under different assumptions, e.g., assuming access to partial causal knowledge [17, 18] or to a randomized trial [16]. Here, we focus on the latter line of research, that is, on answering the following two types of causal questions: *interventional queries*, e.g., "What is the effect of a universal unconditional basic income of 1k EUR on the health of the population?"; and *counterfactual queries*, e.g., "Had Kim received an unconditional basic income of 1k EUR, what would have been the effect on Kim's health?".

Unfortunately, predicting causal effects from observational data alone is in general difficult and often requires strong and impractical causal assumptions. In this context, the *Structural Causal Model* (SCM) [39] is a framework that allows to answer causal queries from observational data, but requires complete causal knowledge. That is, knowledge not only on the parent-children (cause-effect) relationship between every pair of observed variables (i.e., on the causal graph), but also on how these relationships are (i.e., on the structural equations). As a consequence, randomized controlled studies are today still considered to be the gold standard for estimating causal effects. Unfortunately, real world experiments are often expensive to conduct, unethical, or directly impossible.

In this work, we aim at answering the above causal queries, when only observational data and the causal graph are available. Note that the causal graph can often be inferred from domain knowledge [62] or via one of the numerous approaches for causal discovery [27, 54]. We assume causal sufficiency, i.e., that there are no hidden confounders, which are unobserved variables that

affect more than one observed variable. We propose the novel Variational Causal Autoencoder (VCAUSE), a variational graph autoencoder that leverages the causal graph structure and yields accurate estimates of the observational, interventional and counterfactual distributions induced by an unkonwn causal model.

Importantly, we provide the necessary conditions for the design of the encoder and decoder graph neural networks (GNNs), so that the resulting VCAUSE behaves like an SCM. As a result, and without making any assumptions on the true structural equations, VCAUSE provides a practical framework to perform interventions (*do-operator*) and *abduction-action-prediction* steps, which are necessary to evaluate interventional and counterfactual queries.

We evaluate the performance of the proposed VCAUSE model in extensive experiments using observational data from different SCMs, with diverse causal graphs and structural equations. Our experiments show that VCAUSE outperforms competing methods [17, 18] at estimating not only the mean of the interventional/counterfactual distribution, but also the overall distribution, as shown by the quality of its samples (in terms of Maximum Mean Discrepancy, MMD). We finally show a use-case in which VCAUSE is used to assess counterfactual fairness of different classifiers trained on the German Credit dataset [10] as well as to learn accurate and counterfactually fair classifiers.

**Related work.** There are numerous works on causal discovery [15, 18, 27, 30, 33, 40, 49, 54, 56, 58, 60, 63]. In addition, extensive work focuses on interventional and/or counterfactual queries using non-parametric methods [1, 32, 46, 47], and more recently, tractable probabilistic models [59]. Moreover, deep generative models are enjoying increasing attention for causal queries in complex data [31, 35]. Existing approaches focus on estimating the Average Treatment Effect (ATE) by assuming a fixed causal graph that includes the treatment variable [19, 29, 42, 45, 53]; on discovering and intervening on the causal latent structure of the (e.g., image) data [19, 35, 37, 48, 56]; or on addressing interventional and/or counterfactual queries by fitting a conditional model for each observed variable given its causal parents [11, 17, 22, 36, 38]. In the most recent work related to ours [18], the authors propose CAREFL, an autoregressive normalizing flow (ANF) for causal discovery and queries, which focuses on bi-variable scenarios with affine relationships between observed and unobserved variables. In our experiments, we compare VCAUSE with CAREFL (as well as [17]) in more general settings. Finally, up to the best of our knowledge, GNNs have previously been used for causal discovery [58, 61], but have not yet been exploited to address counterfactual and interventional queries, like VCAUSE does.

## 2 Background

In this section, we first provide a brief overview on structural causal models (SCMs) and then introduce the main building block of VCAUSE, i.e., variational graph autoencoders (VGAEs).

### 2.1 Structural causal models

An SCM $\mathcal{M} = (p(\mathbf{U}), \tilde{\mathbf{F}})$ determines how a set of $d$ endogenous (observed) random variables $\mathbf{X} := \{X_1, \ldots X_d\}$ is generated from a set of exogenous (unobserved) random variables $\mathbf{U} := \{U_1, \ldots U_d\}$ (with prior distribution $p(\mathbf{U})$) via the set of *structural equations* $\tilde{\mathbf{F}} = \{X_i := \tilde{f}_i\left(\mathbf{X}_{\mathrm{pa}(i)}, U_i\right)\}_{i=1}^d$. Here $\mathbf{X}_{\mathrm{pa(i)}}$ refers to the set of variables directly causing $X_i$, i.e., parents of $i$. Every SCM $\mathcal{M}$ is associated with a directed acyclic graph (DAG): a *causal graph* $\mathcal{G} := (\mathbf{X}, \mathbf{E})$, for which the nodes (vertices) correspond to endogenous variables $\mathbf{X}$ and the directed edges $\mathbf{E}$ account for the causal parent-child relationship between variables [39]. Given an SCM, there are two types of causal queries of general interest: interventional queries, e.g., "What would happen to the population $\mathbf{X}$, if variable $X_i$ would be set to a fixed value $\alpha$?"; and counterfactual queries, e.g.,"What would have happened to a specific factual sample $\mathbf{x}^F$, had $X_i$ been set to a value $\alpha$?".

More in detail, *interventional queries* aim to evaluate changes in the causal world, or equivalently, manipulations of a subset of the endogenous variables $\mathcal{I} \subseteq [d] := \{1, \ldots, d\}$ at the population level. Interventions on an SCM $\mathcal{M}$ are often represented with the *do-operator* $do(X_i = \alpha_i)$ and lead to a new distribution over the set of endogenous variables $p(\mathbf{X} \mid do(X_i = \alpha_i))$, which is referred to as the *interventional distribution*. In $\mathcal{G}$ an intervention removes incoming edges to node $i$ and sets $X_i = \alpha$ (see Figure 1c). A *counterfactual query* for a given factual instance $\mathbf{x}^F$ aims to estimate what would have happened had $\mathbf{X}_{\mathcal{I}}$ instead taken value $\boldsymbol{\alpha}$. This effect is captured by the

88 *counterfactual distribution* $p(\mathbf{x}^{CF} \mid \mathbf{x}^F, do(X_{\mathcal{I}} = \boldsymbol{\alpha}))$, which can be computed using the abduction-
89 action-prediction approach by Pearl [39]. Refer to Section 3 for further details on the computation of
90 the interventional and counterfactual distributions.

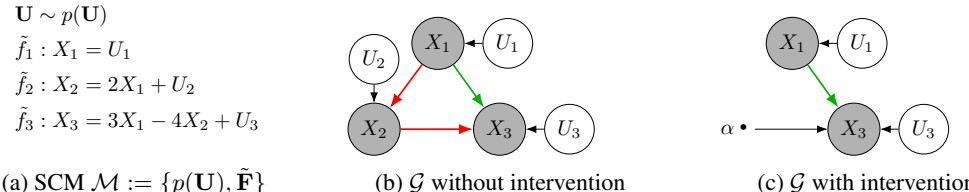

(a) SCM $\mathcal{M} := \{p(\mathbf{U}), \tilde{\mathbf{F}}\}$      (b) $\mathcal{G}$ without intervention      (c) $\mathcal{G}$ with intervention

Figure 1: Example of (a) a *triangle* SCM $\mathcal{M}$ with $d = |\mathbf{X}| = 3$ endogenous variables; (b) correspond-
ing causal graph $\mathcal{G}$ and (c) illustration of an intervention $do(X_2 = \alpha)$ on the causal graph. Green
arrows highlight the direct causal path from $X_1$ to $X_3$, and red arrows the indirect causal path via $X_2$.

## 2.2 Variational Graph Autoencoder and Graph Neural Networks

91

**Variational Autoencoders (VAEs)** [20] are powerful latent variable models based on neural networks
92
(NNs) for jointly i) learning complex and expressive density estimators $p(\mathbf{X}) \approx \int p_\theta(\mathbf{X} \mid \mathbf{Z})p(\mathbf{Z})d\mathbf{Z}$,
93
where the likelihood function (a.k.a. *decoder*) is parameterized using a NN with parameters $\theta$; and ii)
94
performing approximate posterior inference over the latent variables $\mathbf{Z}$ using a variational distribution
95
(a.k.a. *encoder*) $q_\phi(\mathbf{Z} \mid \mathbf{X})$ parameterized using a NN with parameters $\phi$. The parameters $\theta$ and $\phi$
96
are usually learned by maximizing a lower bound on the log-evidence [3, 34, 41, 52].
97

**Variational Graph Autoencoders (VGAEs)** [21] extend VAEs to account for graph-structure in-
98
formation on the data [58]. VGAEs define a (potentially multidimensional) latent variable $Z_i$ per
99
observed variable $X_i$, i.e., $\mathbf{Z} := \{Z_1, \ldots, Z_d\}$. Additionally, VGAEs rely on an adjacency ma-
100
trix $\mathbf{A}$, which is used by two Graph Neural Networks (GNNs), one for the encoder and one for
101
the decoder, to enforce structure on the posterior approximation $q_\phi(\mathbf{Z} \mid \mathbf{X}, \mathbf{A})$ and the likelihood
102
$p_\theta(\mathbf{X} \mid \mathbf{Z}, \mathbf{A})$. More in detail, $\mathbf{A} \in \{0,1\}^{d \times d}$ encodes the graph structure among the observed
103
variables $\mathbf{X} := \{X_1, \ldots X_d\}$, so that $A_{ij} = 1$ if there is a directed edge from $X_j$ to $X_i$, and $A_{ij} = 0$,
104
otherwise. Hence, $\mathbf{A}$ determines which variables $X_i$ influence $Z_j$ $(i, j \in [d])$, and vice versa.
105

**Graph Neural Networks (GNNs)** have generated a lot of attention during the last years, as they
106
achieved significant improvements in graph representation learning [2, 12, 14, 43, 57], While the
107
taxonomy of GNNs is immense [55], in this work we focus on message passing GNNs which allow
108
us to work with directed graphs. In its most general form, a message-passing GNN calculates the
109
output $h_i^l$ for node $i$ in layer $l$ in three steps: i) compute the set of incoming messages arriving to node
110
$i$ from its neighbors $\mathcal{N}_i = \{X_j \mid A_{ij} = 1\}$ using a message function $f^m$ (a NN with parameters $\theta_m^l$),
111
that is $\{m_{ij}^l\}_{j \in \mathcal{N}_i} = \{f_i^m(h_i^{l-1}, h_j^{l-1}; \theta_m) \mid j \in \mathcal{N}_i\}$ ; ii) combine the set of messages into a single
112
message $M_i^l := f^a(\{m_{ij}^l\}_j)$ using an aggregation function $f^a$ (e.g. adding up the messages); and iii)
113
update the node state $h_i^l := f^u(h_i^{l-1}, M_i^l; \theta_u^l)$, using an update function $f^u$ (a NN with parameters
114
$\theta_u^l$). As a result, the output $h_i^l$ can be written as
115

$$h_i^l = f^u \left( h_i^{l-1}, f^a \left( \{f^m(h_i^{l-1}, h_j^{l-1}; \theta_m^l) \mid j \in \mathcal{N}_i\} \right); \theta_u^l \right). \tag{1}$$

Note that the above expression assures that *the output for each node $i$ is computed using information*
116
*from its neighbors $\mathcal{N}_i$ according to* $\mathbf{A}$. Moreover, if the GNN has $N_h$ hidden layers, then the
117
output for node $i$ not only depends on its direct neighbors $\mathcal{N}_i$, but also on its neighbors up to order
118
$N_h + 1$ (hops). As an example, if $N_h = 0$ then the output for each node only depend on its direct
119
neighbors (*parents*). If instead $N_h = 1$, then the output for each node depends on 2-hop neighbors
120
(*grand-parents*). For a detailed description of GNNs, please refer to Appendix A.
121

## 3 Observational, interventional and counterfactual distributions

122

In this section, we introduce the observational, interventional and counterfactual distributions (trig-
123
gered by any intervention $do(\mathbf{X}_{\mathcal{I}} = \boldsymbol{\alpha})$) that are induced from an SCM $\mathcal{M} := \{p(\mathbf{U}), \tilde{\mathbf{F}}\}$. Specifi-
124
cally, we summarize the main properties of an SCM that will allow us to propose a novel class of
125

VGAEs, the variational causal autoencoders (VCAUSE), to compute accurate estimates of these distributions using observational data and a known causal graph. To this end, we assume the absence of hidden confounders, i.e., we assume that $p(\mathbf{U}) = \prod_{i=1}^{d} p(U_i)$.

**Observational distribution.** The SCM $\mathcal{M}$ determines the observational distribution $p(\mathbf{X})$ over the set of endogenous variables $\mathbf{X} = \{X_1, \ldots X_d\}$, which satisfies causal factorization [44], i.e., $p(\mathbf{X}) = \prod_{i=1}^{d} p(X_i \mid \mathbf{X}_{\mathrm{pa}(i)})$. That is, after marginalizing out the exogenous variables $\mathbf{U}$, the distribution of each endogenous variable $X_i$ depends only on its parents, i.e., $\mathbf{X}_{\mathrm{pa}(i)}$. The *observational distribution* can alternatively be written only in terms of the exogenous variables $\mathbf{U}$ as

$$p(\mathbf{X}) = \int \mathbf{F}(\mathbf{U}) p(\mathbf{U}) d\mathbf{U}, \tag{2}$$

where $\mathbf{F} : \mathbf{U} \to \mathbf{X}$ corresponds to the set of structural equations, equivalent to $\tilde{\mathbf{F}}$, that directly transform the exogenous variables $\mathbf{U}$ into the endogenous variables $\mathbf{X}$. Let us denote by $an(i)$ the set of indexes of the ancestors of $i$, and $an^*(i) := an(i) \cup \{i\}$. Then, the causal factorization induced by the SCM $\mathcal{M}$ leads to the following property of $\mathbf{F}(\mathbf{U})$:

**Property 1.** *Each endogenous variable $X_i$ can be expressed as a function of its exogenous variable $U_i$ and the ones of all its causal ancestors, i.e., $\mathbf{F}(\mathbf{U}) := \{X_i = f_i(\{U_j \mid j \in an^*(i)\})\}$. This, together with the causal sufficiency assumption, implies that $X_i$ is statistically independent of $U_j, \forall j \notin an^*(i)$.*

**Interventional distribution.** As stated in Section 2.1, interventions on a set of variables $\mathcal{I}$ can be performed using the *do-operator*, which can be seen as a mapping $do(\mathbf{X}_{\mathcal{I}} = \boldsymbol{\alpha}) : \mathcal{M} \mapsto \mathcal{M}^{\mathcal{I}} = (p(\mathbf{U}), \tilde{\mathbf{F}}^{\mathcal{I}})$ where $\tilde{\mathbf{F}}^{\mathcal{I}} = \{\tilde{f}_j \mid j \notin \mathcal{I}\} \cup \{\alpha_i \mid i \in \mathcal{I}\}$. As above, we can represent the resulting set of *intervened structural equations* $\mathbf{F}^{\mathcal{I}} = \{f_j \mid j \notin \mathcal{I}\} \cup \{\alpha_i \mid i \in \mathcal{I}\}$ in terms of only the exogenous variables $\mathbf{U}$, so that we can write the *interventional distribution* as:

$$p(\mathbf{X} \mid do(\mathbf{X}_{\mathcal{I}} = \boldsymbol{\alpha})) = \int \mathbf{F}^{\mathcal{I}}(\mathbf{U}) p(\mathbf{U}) d\mathbf{U}. \tag{3}$$

Assuming an intervention $do(\mathbf{X}_{\mathcal{I}} = \boldsymbol{\alpha})$ on $\mathcal{M}$, then the resulting structural equations $\mathbf{F}^{\mathcal{I}}(\mathbf{U})$ satisfy:

**Property 2.** *After an intervention $do(\mathbf{X}_{\mathcal{I}} = \boldsymbol{\alpha})$ on $\mathcal{M}$, all the causal paths from $U_j$ $\forall j \in an^*(i)$ to $X_i$ that include an intervened variable in $\mathbf{X}_{\mathcal{I}}$ (i.e., the causal paths where $\mathbf{X}_{\mathcal{I}}$ is a mediator) are severed in $\mathbf{F}^{\mathcal{I}}$, while the rest of causal paths remain untouched.*

The above property is illustrated in Figure 1, where we can easily observe that after an intervention $do(X_2 = \alpha)$, the indirect causal path (in red) from $X_1$, and thus from $U_1$, to $X_3$ via $X_2$ is severed, while the direct path (in green) remains.

**Counterfactual distribution.** Assuming the SCM $\mathcal{M} = \{p(\mathbf{U}), \tilde{\mathbf{F}}\}$ to be known, the following three steps defined by Pearl [39] allow us to compute counterfactuals $\mathbf{x}^{CF}$ as: i) *Abduction:* infer the values of the exogenous variables $\mathbf{U}$ for a factual sample $\mathbf{x}^F$, i.e., compute $p(\mathbf{U} \mid \mathbf{x}^F)$; ii) *Action:* intervene with $do(\mathbf{X}_{\mathcal{I}} = \boldsymbol{\alpha}) : \mathcal{M} \mapsto \mathcal{M}^{\mathcal{I}} = (p(\mathbf{U}), \tilde{\mathbf{F}}^{\mathcal{I}})$; and iii) *Prediction:* use the posterior distribution $p(\mathbf{U} \mid \mathbf{x}^F)$ and the new structural equations $\tilde{\mathbf{F}}^{\mathcal{I}}$ to compute $p(\mathbf{x}^{CF} \mid \mathbf{x}^F)$. The prediction step can be alternatively computed using the new set of structural equations $\mathbf{F}^{\mathcal{I}}$ defined in terms of the exogenous variables $\mathbf{U}$, so that we can write the *counterfactual distribution* as:

$$p(\mathbf{x}^{CF} \mid \mathbf{x}^F, do(\mathbf{X}_{\mathcal{I}} = \boldsymbol{\alpha})) = \int \mathbf{F}^{\mathcal{I}}(\mathbf{U}) p(\mathbf{U} \mid \mathbf{x}^F) d\mathbf{U}. \tag{4}$$

Importantly, the resulting posterior distribution $p(\mathbf{U} \mid \mathbf{x}^F)$ satisfies:

**Property 3.** *In the abduction step, statistical independence implies that conditioned on the endogenous variables of the factual sample $\mathbf{x}^F$, each exogenous variable $U_i$ is independent of the factual value $x_j^F$ if $j \neq i$ and the variable $X_j$ is not a parent of $X_i$, i.e., $j \notin pa^*(i) := pa(i) \cup \{i\}$.*

## 4 Variational Causal Autoencoder (VCAUSE)

In this section, we present a novel variational causal graph autoencoder (VCAUSE) to approximate the observational, interventional and counterfactual distributions given in (2), (3) and (4), respectively.

168 While the underlying SCM $\mathcal{M}$ is unknown, we assume access to: the causal graph $\mathcal{G}$ and observational
169 data $\{\mathbf{x}_n\}_{n=1}^N$, i.e., i.i.d. samples of the observational distribution induced by $\mathcal{M}$.

170 **Definition 4.1.** *(VCAUSE)* . Given a causal graph $\mathcal{G}$ over a set of endogenous variables $\mathbf{X} =$
171 $\{X_1, \ldots, X_d\}$, which establishes the set of parents $pa^*(i)$ for each variable $X_i$ (including the $i$-th
172 node). A variational causal graph autoencoder (VCAUSE) is defined by:

- 173 • A causal adjacency matrix $\mathbf{A}$, which is a $d \times d$ binary matrix with elements $A_{ij} = 1$ if
  174 $j \in pa^*(i)$, i.e., when $i = j$ or $j$ is a parent of $i$. Otherwise, $A_{ij} = 0$.
- 175 • A prior distribution $p(\mathbf{Z}) = \prod_i p(Z_i)$ over the set of latent variables $\mathbf{Z} = \{Z_1, \ldots, Z_d\}$.
- 176 • A decoder $p_\theta(\mathbf{X} \mid \mathbf{Z}, \mathbf{A})$, which is a GNN (parameterized by $\theta$) that takes as input the set
  177 of latent variables $\mathbf{Z}$ and the causal adjacency matrix $\mathbf{A}$, and outputs the parameters of the
  178 likelihood $p_\theta(\mathbf{X} \mid \mathbf{Z}, \mathbf{A})$.
- 179 • An encoder $q_\phi(\mathbf{Z} \mid \mathbf{X}, \mathbf{A})$, which is a GNN (parameterized by $\phi$) that takes as input the
  180 endogenous variables $\mathbf{X}$ and the causal adjacency matrix $\mathbf{A}$, and outputs the parameters of
  181 the posterior approximation $q_\phi(\mathbf{Z} \mid \mathbf{X}, \mathbf{A})$.

182 Given observational data $\{\mathbf{x}_n\}_{n=1}^N$, one may learn the parameters $\theta$ and $\phi$ that best estimate the
183 density $p(\mathbf{X})$. We here rely on the partially importance weighted auto-encoder (PIWAE) [41].

184 Next, we discuss how to design VCAUSE such that it is able to capture the observational, inter-
185 ventional, and counterfactual distribution induced by an unknown SCM. Importantly, we derive the
186 necessary conditions on the design of both the encoder and decoder GNNs such that VCAUSE fulfills
187 the SCM properties introduced in Section 3.

## 4.1 Observational distribution

189 VCAUSE approximates the *observational distribution* in (2) using the generative model as

$$p(\mathbf{X}) \approx \int p_\theta(\mathbf{X} \mid \mathbf{Z}, \mathbf{A})p(\mathbf{Z})d\mathbf{Z} = \int \prod_{i=1}^d p_\theta(X_i \mid \mathbf{Z}, \mathbf{A})p(\mathbf{Z})d\mathbf{Z}. \tag{5}$$

190 Figure 2a depicts this generative process. If we compare (5) with the true observational distribution
191 in (2), we observe that the latent variables $\mathbf{Z}$ play a similar role to the exogenous variables $\mathbf{U}$,
192 and the decoder $p_\theta(\mathbf{X} \mid \mathbf{Z}, \mathbf{A})$ plays a similar role to the structural equations $\mathbf{F}$. Yet, we remark
193 that $\mathbf{Z}$ does not need to correspond to the exogenous variables, i.e., $p(\mathbf{U}) \neq p(\mathbf{Z})$, in order for (5)
194 to provide a good approximation of the observational distribution in (2). In fact, standard VAEs
195 perform accurate density estimation using observational data, without the need for capturing causal
196 information. However, in this paper, we seek to ensure that our observational distribution induced
197 by VCAUSE complies causal factorization (**Property** 1). To that end, we need to make sure that
198 $p_\theta(X_i \mid \mathbf{Z}, \mathbf{A}) = p_\theta(X_i \mid \mathbf{Z}_{an^*(i)})$. That is, $X_i$ depends only on $Z_j$ if $j = i$ or $X_j$ is an ancestor of
199 $X_i$ in the causal graph. To fulfill this property, the GNN of the decoder should satisfy the following:

200 **Proposition 1.** *(Causal factorization). VCAUSE satisfies causal factorization, $p_\theta(\mathbf{X} \mid \mathbf{Z}, \mathbf{A}) =$*
201 $\prod_i p_{\theta_i}(X_i \mid \mathbf{Z}_{an^*(i)})$*, if and only if the number of hidden layers in the decoder is greater or equal*
202 *than $\delta - 1$, with $\delta$ being the longest shortest directed path between any two endogenous nodes.*

203 The above proposition (proved in Appendix B) is based on the fact that, in a GNN with $N_h$ hidden
204 layers (and $N_h + 1$ layers in total), the output for the $i$-th node depends on its neighbors of up
205 to $N_h + 1$ hops. As an example, consider the following *chain* causal graph: $X_1 \rightarrow X_2 \rightarrow X_3$,
206 such that $\delta = 2$. In the decoder, the first layer yields a hidden representation for the 3-rd node
207 $h_3^1 := f(f(Z_2), Z_3)$ that only depends on $Z_2$ and $Z_3$. Thus, we need a second layer for its output
208 $h_3^2 := f(h_2, Z_3) = f(f(f(Z_1), Z_2), Z_3)$ to depend on $Z_1$ (note that $X_1$ is an ancestor of $X_3$).

## 4.2 Interventional distribution

210 VCAUSE approximates the *interventional distribution* in (3) as (illustrated Figure 3):

$$p(\mathbf{X} \mid do(\mathbf{X}_\mathcal{I} = \boldsymbol{\alpha})) \approx \int p_\theta(\mathbf{X} \mid \{Z_i\}_{i \notin \mathcal{I}}, \{Z_i^\mathcal{I}\}_{i \in \mathcal{I}}, \mathbf{A}^\mathcal{I})p(\mathbf{Z})q_\phi(\mathbf{Z}^\mathcal{I} \mid \mathbf{A}^\mathcal{I}, \mathbf{X}_\mathcal{I})d\mathbf{Z}, \tag{6}$$

211 where the *do-operator* is performed on the causal adjacency matrix as $do(\mathbf{X}_\mathcal{I} = \boldsymbol{\alpha}) : \mathbf{A} \mapsto \mathbf{A}^\mathcal{I} =$
212 $\{A_{ij}\}_{\forall i \notin \mathcal{I}, j} \cup \{A_{ij} = 0\}_{\forall i \in \mathcal{I}, j}$. This ensures that $X_i$ for $i \in \mathcal{I}$ is independent of $Z_j$ for all

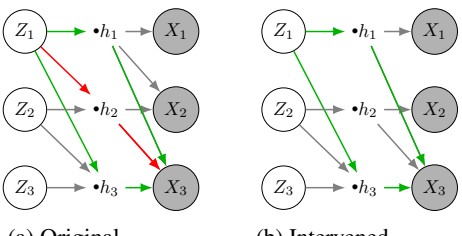

Figure 2: VCAUSE generation of (a) observational, (b) interventional, and (c) counterfactual samples. The 'hat' in $\hat{\mathbf{X}}$ and $\hat{\mathbf{x}}^{CF}$ indicate that they are sample estimates of the true random variables.

$j \neq i$. Note that in order for (6) to be able to approximate the interventional distribution in (3), an intervention on a variational causal autoencoder should satisfy **Property 2**, i.e.:

**Proposition 2.** *(Causal interventions). VCAUSE can capture causal interventions if and only if the number of hidden layers in its decoder is greater than or equal to $\gamma - 1$, with $\gamma$ being the longest directed path between any two endogenous nodes in $\mathcal{G}$.*

To illustrate this, Figure 3 depicts how messages are exchanged in a one-hidden-layer decoder GNN corresponding to the causal graph $\mathcal{G}$ in Figure 1 (*triangle* with $\gamma = 2$), both (a) without and (b) with an intervention on $X_2$. We highlight in green the direct messages (sent via direct causal path in $\mathcal{G}$), and in red the indirect messages (sent via indirect causal path in $\mathcal{G}$) from $Z_1$ to $X_3$. Observe that, similarly to Figure 1, in (a) there is an indirect path (via $h_2$) from $Z_1$ to $X_3$; while in (b) this path is severed. Hence, the hidden layer $(h_1, h_2, h_3)$ allows to differentiate between direct and indirect paths and thus to capture interventional effects.

(a) Original    (b) Intervened

Figure 3: VCAUSE decoder (a) with and (b) without intervening on $X_2$. Arrows indicate message passing in the GNN corresponding to direct (green) and indirect (red) causal paths in Figure (1).

As the condition in Proposition 2 is more restrictive than the one in Proposition 1, in order for VCAUSE to be able to capture observational and interventional distributions, it should satisfy that:

**Design condition 1:** *The decoder GNN of VCAUSE has at least as many hidden layers as $\gamma - 1$, with $\gamma$ being the longest directed path in the causal graph $\mathcal{G}$.*

### 4.3 Counterfactual distribution

VCAUSE approximates the *counterfactual distribution* in (4) as (illustrated in Figure 2c):

$$p(\mathbf{x}^{CF} \mid do(\mathbf{X}_\mathcal{I} = \boldsymbol{\alpha}), \mathbf{x}^F) \approx \int \underbrace{\underbrace{p_\theta(\mathbf{X} \mid \{Z_i^F\}_{i \notin \mathcal{I}}, \{Z_i^\mathcal{I}\}_{i \in \mathcal{I}}, \mathbf{A}^\mathcal{I}) q_\phi(\mathbf{Z}^\mathcal{I} \mid \mathbf{x}^\mathcal{I}, \mathbf{A}^\mathcal{I})}_{action} \underbrace{q_\phi(\mathbf{Z}^F \mid \mathbf{x}^F, \mathbf{A})}_{abduction}}_{prediction} d\mathbf{Z},$$

where $\mathbf{x}^F$ represents a sample from $\mathbf{X}$ for which we seek to compute the distribution over counterfactual $\mathbf{x}^{CF}$. Note here that two different passes of the encoder are necessary: one for the *abduction* step of the factual instance $q_\phi(\mathbf{Z}^F \mid \mathbf{x}^F, \mathbf{A})$; and another one for the *action* step (intervention) $q_\phi(\mathbf{Z}^\mathcal{I} \mid \mathbf{x}^\mathcal{I}, \mathbf{A}^\mathcal{I})$ with $x_i^\mathcal{I} = \alpha_i \ \forall i \in \mathcal{I}$ (we remark that the rest of the values in $\mathbf{x}^\mathcal{I}$ do not affect the overall counterfactual computation). We then evaluate the likelihood, making sure that the resulting counterfactual sample $\mathbf{x}^{CF}$ only depends on the $\{Z_i^F\}_{i \notin \mathcal{I}} \subseteq \mathbf{Z}^F$ and $\{Z_i^\mathcal{I}\}_{i \in \mathcal{I}} \subseteq \mathbf{Z}^\mathcal{I}$. Importantly, in order for VCAUSE to be able to approximate the counterfactual distribution, we need its abduction (and action) step(s) to comply with **Property 3**, i.e.:

**Proposition 3.** *(Abduction). The abduction step of an observed sample $\mathbf{x} = \{x_1, \ldots, x_d\}$ in a variational causal autoencoder satisfies that for all $i$ the posterior of $Z_i$ is independent on the subset $\{x_j\}_{j \notin pa^*(i)} \subseteq \mathbf{x}$, if and only if the encoder GNN has no hidden layers.*

The above result (proved in Appendix B) can be shown by the message passing algorithm computed by the encoder GNN, and leads to the second condition that VCAUSE should satisfy by design:

Table 1: Evaluation of the observational and interventional distributions generated by VCAUSE with different numbers of hidden layers $N_h$. All metrics are shown in percentage (%).

| $N_h$ | collider ($\delta = 1, \gamma = 1$) | | triangle ($\delta = 1, \gamma = 2$) | | chain ($\delta = 2, \gamma = 2$) | |
|---|---|---|---|---|---|---|
| | MMD Obs. (%) | MMD Inter.(%) | MMD Obs.(%) | MMD Inter.(%) | MMD Obs.(%) | MMD Inter.(%) |
| 0 | $1.37 \pm 0.54$ | $0.90 \pm 0.19$ | $2.20 \pm 0.74$ | $4.03 \pm 0.42$ | $5.58 \pm 1.01$ | $8.07 \pm 0.53$ |
| 1 | $0.86 \pm 0.34$ | $0.95 \pm 0.28$ | $1.05 \pm 0.38$ | $2.35 \pm 0.35$ | $1.4 \pm 0.31$ | $1.56 \pm 0.4$ |
| 2 | $1.0 \pm 0.50$ | $0.91 \pm 0.16$ | $1.20 \pm 0.63$ | $2.33 \pm 0.29$ | $1.67 \pm 0.61$ | $1.46 \pm 0.29$ |

**Design condition 2:** *The encoder GNN of VCAUSE has no hidden layers.*

Note that while the above condition may look restrictive and limiting the capacity of our encoder, we may choose arbitrarily complex NNs to model the message $f^m$ and update $f^u$ functions, as well as one or more aggregation functions $f^a$, e.g., sum, mean or max, to model the encoder [8].

### 4.4 Practical considerations

Next, we briefly discuss practical implementation considerations to handle complex causal models, which often appear in real world applications–see the causal graph of the German Credit dataset [10] in Section 6 for an example. For further details on VCAUSE implementation, refer to Appendix C.

**Heterogeneous endogenous variables:** In general GNNs are parametrized such that the parameters of the message $f^m$ and update $f^u$ functions are shared for all the nodes and edges in the graph. However, similarly as in the structural causal equations $\mathbf{F}$, we can define a different message function $f_{ij}^m$ for every edge in the causal graph by assuming a different set of parameters $\theta_{mij}$ per edge in (1). Similarly, we can also assume different update functions $f_i^u$ for each node $i$, by considering different update parameters $\theta_{ui}$ for each node. This allows us to use different functions for each node, and thus model heterogeneous endogenous variables, in terms of their continuous/discrete distribution, and also of their structural equations, e.g., linear/non-linear.

**Heterogenous causal nodes:** So far, we have modeled each endogenous variable $X_i$ as a node in the causal graph $\mathcal{G}$, and thus in the VCAUSE GNNs. However, in some application domains the relationships between a subset of variables may be unknown, or they may be affected by hidden confounders, leading to an undirected path between them. In such cases, the subset of $(k_i)$ variables is modeled as a multidimensional and potentially heterogeneous node $\mathbf{X}_i = \{X_{i1}, \ldots, X_{ik_i}\}$. Note that all the variables in the multidimensional node $\mathbf{X}_i$ share the same latent random variable $Z_i$.

## 5 Evaluation

In this section, we conduct extensive experiments to evaluate the performance of VCAUSE at estimating the outcomes of causal queries. Please refer to Appendix D for a complete description of the experimental set-up. Moreover, to ease the reproducibility of our experiments, our code is publicly available at https://github.com/XXXX/XXXXX.

**Datasets.** We consider different synthetic causal graphs that differ in the number of nodes $d$, diameter $\delta$, and longest path $\gamma$: synthetic *collider* ($d = 3, \delta = 1, \gamma = 1$), *M-graph* ($d = 3, \delta = 1, \gamma = 1$), *triangle* ($d = 3, \delta = 1, \gamma = 2$), *chain* ($d = 3, \delta = 2, \gamma = 2$), and a semi-synthetic *loan* ($d = 7, \delta = 2, \gamma = 3$) from [17]. For all of the synthetic datasets (i.e., except *loan*), we consider three different types of structural equations with increasing complexity: linear additive noise (LIN), non-linear additive noise (NLIN) and non-additive noise (NADD).

**Metrics.** We evaluate the observational distribution using the Maximum Mean Discrepancy (MMD) [13] as distance-measure between the true and estimated distributions as a whole, i.e., the lower the MMD the better the distributions match. For the interventional distribution, we additionally report the estimation squared error for the mean and for the standard deviation (MeanE and StdE respectively) for the children of the intervened variables. For the counterfactual distribution we report the mean square error (MSE) as well as the standard deviation of the squared error (SSE) between the true and the estimated counterfactual value. We compute all results over 10 independent runs.

**Validating VCAUSE design conditions.** In a first step we empirically validate our design choices for the VCAUSE encoder and decoder. We show how the number of hidden layers $N_h$ in the decoder affect the quality of the estimation of the observational and interventional distributions for three

Table 2: Performance of different methods at estimating the observational, interventional and counterfactual of different SCMs. All metrics are shown in percentage (%).

| SCM | | Model | Obs. MMD (%) | Interventional MMD (%) | MeanE (%) | StdE (%) | Counterfactuals MSE (%) | SSE (%) |
|---|---|---|---|---|---|---|---|---|
| triangle | LIN | MultiCVAE | **1.07±0.88** | 4.92±2.00 | 0.81±0.33 | **24.39±0.20** | 15.52±4.69 | 12.78±5.07 |
| | | CAREFL | 5.51±0.80 | 3.63±0.22 | **0.18±0.05** | 50.10±0.79 | **5.11±0.87** | **6.18±0.81** |
| | | VCAUSE | **1.26±0.68** | **2.21±0.26** | 0.65±0.12 | 24.51±0.09 | 11.68±0.69 | 7.62±0.42 |
| | NLIN | MultiCVAE | **1.15±0.83** | 7.21±3.90 | **0.57±0.29** | **17.58±0.26** | 12.92±4.11 | 10.03±5.33 |
| | | CAREFL | 5.37±1.18 | 8.15±0.76 | 1.14±0.38 | 60.48±1.36 | **8.03±1.53** | 8.95±1.42 |
| | | VCAUSE | **1.55±0.90** | **6.26±1.31** | 0.85±0.16 | 17.41±0.09 | 12.10±0.95 | **8.17±0.64** |
| | NADD | MultiCVAE | **2.15±0.58** | 43.63±2.73 | 0.18±0.07 | **19.14±1.75** | 24.45±1.62 | 38.23±3.83 |
| | | CAREFL | 6.14±1.33 | 76.84±14.78 | 2.59±3.76 | 112.65±6.08 | **8.32±0.93** | 39.82±0.88 |
| | | VCAUSE | **2.54±1.18** | **8.87±1.52** | **0.09±0.04** | 20.94±1.72 | 10.36±0.78 | **17.82±1.20** |
| loan | | MultiCVAE | 76.18±12.61 | 188.35±9.05 | 16.84±5.64 | 60.29±3.39 | 72.41±4.75 | 38.69±1.16 |
| | | CAREFL | 9.28±2.15 | 9.54±1.82 | 3.55±2.48 | 28.94±1.15 | 32.54±0.21 | 17.68±0.34 |
| | | VCAUSE | **1.09±0.24** | **1.41±0.16** | **0.40±0.09** | **9.58±0.06** | **30.06±0.14** | **14.22±0.11** |

SCMs, with different values of longest shortest directed path $\delta$ and longest directed path $\gamma$. In Table 1, we observe that as expected: i) the *collider* ($\delta = \gamma = 1$) does not need any hidden layer to provide accurate estimate of both the observational and interventional distributions. In contrast, the *triangle* ($\delta = 1, \gamma = 2$), which according to Proposition 2 needs at least one hidden layer to get a more accurate estimate of the interventional distribution (while an improvement in the observational is not as evident). Finally, as stated by Propositions 1 and 2, the *chain* ($\delta = \gamma = 2$) requires at least one hidden layer to accurately approximate both the observational and interventional distributions.

## 5.1 Estimating interventional and counterfactual distributions

In the following we evaluate the potential of VCAUSE to model interventional and counterfactual queries. We consider interventions of the form $do(x_i = \alpha_i)$ for several values of $\alpha_i$ on both root and non-root nodes. Here we report the results for the *triangle* and *loan* graphs. Refer to Appendix E for the remaining results.

**Baselines.** We compare our VCAUSE with two competing methods: i) MultiCVAE, which trains a conditional VAE for each endogenous variable that is not a root node in the causal graph [17]; and ii) CAREFL [18], which relies on autoregressive causal flows to estimate counterfactual queries.

**Results for interventional distributions.** Table 2 (middle columns) reports the MMD, MeanE, and StdE for the interventional distribution. Here we can observe that VCAUSE consistently outperforms other methods in terms of MMD. Note that the three methods provide comparable results in capturing the mean of the interventional distribution (MeanE) (except for the more complex *loan* graph, where VCAUSE outperforms the others). However, it can also be seen that CAREFL and MultiCVAE often fail to capture the standard deviation of the interventional distribution (StdE), while VCAUSE provides a more accurate estimate of the overall interventional distribution.

**Results for the counterfactuals.** Table 2 also reports the results for the counterfactual distribution. Here, we first observe that MultiC-VAE slightly underperforms the other two models. Second, we observe that CAREFL provides more accurate estimates than VCAUSE in terms of MSE, which may be explained by the fact that CAREFL performs exact inference. However, CAREFL presents high variance in its results (see SSE). Note that to perform interventions, CAREFL sets the parents of the intervened vari-

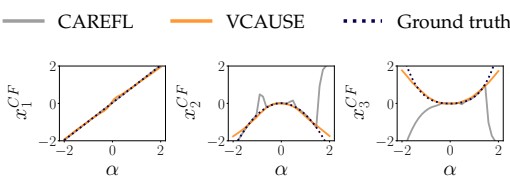

Figure 4: Example of counterfactuals for a factual $\mathbf{x}^F$ from the test set of the *triangle* NLIN and $do(x_1 = \alpha)$.

ables to zero, which may not completely severe the causal paths to the intervened nodes. In contrast, as further illustrated in Figure 4, VCAUSE leads to consistent counterfactual estimations across factual samples and interventions. Figure 4 also shows that CAREFL fails severely for some intervention values, despite of intervening on a root node.

## 6 Use case: counterfactual fairness

Finally, we showcase the practical use of VCAUSE for assessing counterfactual fairness and also for training a counterfactually fair classifier. To this end, we use the German Credit dataset publicly available at the UCI repository [50]. We rely on the causal model with the following random variables $\mathbf{X}$ as proposed in [6] (see Figure 5): sensitive feature $S = \{sex\}$, and non-sensitive features $C = \{age\}$, $R = \{credit\ amount, repayment\ history\}$ and $H = \{checking\ account, savings, housing\}$. Then, we aim to predict the binary feature $Y = \{credit\ risk\}$ from $\mathbf{X}$. See Appendix F for further details.

**Counterfactual fairness.** Let $S \subset \mathbf{X}$ be a sensitive attribute (e.g., gender), then a classifier $h : \mathbf{X} \rightarrow Y$ is considered $\epsilon$-counterfactually fair [24] if:

$$\left| P(h(\mathbf{x}^{CF}) = y \mid do(S = \alpha), \mathbf{x}^F) - P(h(\mathbf{x}^{CF}) = y \mid do(S = \alpha'), \mathbf{x}^F) \right| \leq \epsilon, \quad \forall \mathbf{x}^{CF}, \alpha' \neq \alpha, y.$$

A classifier is counterfactually fair ($\epsilon = 0$), if, given a factual $\mathbf{x}^F$ with sensitive attribute $S = \alpha$, had its sensitive attribute been different $S = \alpha'$, the classifier prediction would remain the same. As VCAUSE allows us to generate counterfactual samples, we can thus use it to *audit* the fairness level of a classifier. Moreover, we can use the VCAUSE encoder to *learn a fair classifier* $h_{\mathrm{VCAUSE}} : \mathbf{Z} \backslash Z_S \rightarrow Y$, which takes as input the latent variables generated by VCAUSE without the one of the sensitive attribute $Z_S$. Following [24], we compare our VCAUSE fair classifier $h_{\mathrm{VCAUSE}}$ with: i) a *full* model $h_{\mathrm{full}} : \mathbf{X} \rightarrow Y$ that takes as input the complete variable set; ii) an *unaware* model $h_{\mathrm{unaw}} : \mathbf{X} \backslash S \rightarrow Y$ that takes as input all variables but the sensitive one; iii) and a *fair* model $h_{\mathrm{fair}} : \{X_i | S \not\in an^*(i)\} \rightarrow Y$ that takes as input all non-descendant variables of the sensitive attribute.

**Results.** The results for logistic regression (LR) and support vector machine (SVM) classifiers are summarized in Table 3. Note that VCAUSE correctly ranks the different methods based on their unfairness level, showing that the *full* classifier is consistently less fair than the *unaware* and the *fair* classifiers, respectively. Moreover, the VCAUSE classifier leads to a fair classifier, while keeping the f1-score comparable to the unfair classifier. Therefore, VCAUSE does not only allow us to audit counterfactual fairness but also provides a practical approach to train accurate and fair classifiers.

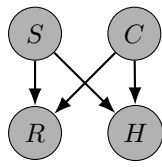

Figure 5: Causal graph for variables $\mathbf{X}$ of the German Credit dataset [6].

Table 3: Evaluation of counterfactual (un)fairness. All metrics are shown in %. Lower/Larger values of unfairness/f1-score are better.

| Metric | Classifier | full | unaware | fair | VCAUSE |
|---|---|---|---|---|---|
| ↑ f1-score (%) | LR | 71.07 | 68.33 | 50.00 | 74.81 |
| | SVM | 74.60 | 72.44 | 64.71 | 70.40 |
| ↓ unfairness (%) | LR | 5.93 | 2.25 | 0.16 | 0.85 |
| | SVM | 6.07 | 2.68 | 0.20 | 1.00 |

## 7 Conclusion

In this work, we have proposed VCAUSE a variational causal autoencoder based on GNNs that: i) is specially designed to capture the properties of SCMs; ii) inherently handles heterogeneous causal graphs and data; and iii) provides accurate estimates of interventional and counterfactual distributions for SCMs of different complexities. As demonstrated by extensive experiments, VCAUSE provides accurate results for a wide variety of interventions in diverse SCMs leading to significantly more robust results than competing methods [17, 18]. Finally, we have shown a practical use-case of VCAUSE in a problem of increasing interest for the machine learning community, namely, fairness in classification. In particular, we have shown how to use VCAUSE to both assess counterfactual fairness and to train counterfactually fair classifiers.

Moreover, our work opens up many interesting venues for future work. First, as we have assumed a known causal graph and the absence of hidden confounders, it would be important to evaluate the sensitivity of VCAUSE to the violation of these assumptions in order to avoid its misuse. We also plan to extend VCAUSE to handle hidden confounders and to perform efficient causal discovery. Second, it would be interesting to perform ablation studies on the limitations of available GNNs architectures [55] for the VCAUSE encoder and decoder; as well as on how the performance of GNNs deteriorates as we increase the length of the causal path and thus the required number of hidden layers [28]. Finally, it would be interesting to apply VCAUSE to other causal questions recently discussed in the machine learning literature, such as privacy-preserving causal inference [26] or explainable machine learning [17].

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
