# OpenReview forum: "Variational Causal Autoencoder for Interventional and Counterfactual Queries"
_NeurIPS.cc/2021/Conference — NeurIPS 2021 Submitted_

### Official Review · Reviewer_oFSi · 2021-07-11

**Rating:** 4
**Confidence:** 4

**Summary:**

The paper introduces Variational Causal Autoencoder (VCAUSE) for causal inference in the absence of hidden variables. It is designed to capture the properties of SCM.


**Limitations And Societal Impact:**

Yes

**Main Review:**

Strengths. The paper is well written.

Weaknesses. The authors state that the VCAUSE was developed for causal inference, however, they also state that they assumed that the causal graph is known in the current contribution. If it is the case, why there is the need for the causal inference?

Although the paper is quite clearly written, there are some flows. First, the role of U (exogenous) variables is not clear: is is assumed that there is 1 latent variable per 1 observed variable, as shown on Figure 1. Then, these U variables are not mentioned further in the paper. The VCAUSE (Definition 4.1) is defined without them. It is stated that "the latent variables Z play a similar role to the exogenous variables U" but it is very vague, and I do not see a clear correspondence.

Propositions 1 and 2 are important but I guess no novel, it should be a known result in deep learning.

I would appreciate more explanations on the design condition 2. I see that it follows from Prop. 2 but it is unclear why to use neural networks without hidden layers.


**Time Spent Reviewing:**

4

---

> ### Author Response · Authors · 2021-08-10
> **Reply to Reviewer oFSi**
>
> **Causal inference versus causal discovery.** We believe there has been a misunderstanding. Our model is based on the Structural Causal Model (SCM) framework. Here, causal discovery deals with finding the relationships between the features (i.e., finding the causal graph), while causal inference deals with finding or approximating how they relate, i.e., the structural causal equations (SEM). They are two completely different tasks. Our focus, as stated in the abstract and repeatedly in the main manuscript, e.g., Line 32, is (approximate) causal inference. To do so, we need more than observational data. In our case, we  assume we know the causal graph (abstract line 3). We could relatex our assumption by inferring the causal graph using any of the causal discovery approaches already cited in the related work section of the paper.
>
> **U versus Z.** Let us try to further clarify the difference between the true exogenous variables $U$ and the latent variable of VCAUSE $Z$. We remark that the latent variables $Z$ play a similar role to the exogenous variables $U$, however we emphasize that $Z$ does not need to correspond to the exogenous variables. As the structural equations and the true distribution of the exogenous variables $U$ is unknown, there is no way we can ensure that VCAUSE latent variables $Z$ correspond to the true exogenous variables. That said, what we assume is that there is one independent latent variable $Z_i$ for every observed variable $X_i$ capturing all the information of $X_i$ that cannot be explained by its parents. Thus, since $X_i$ is in turn a (deterministic) function of its parent and its exogenous variable $U_i$, the encoded $Z_i$ (actually its posterior distribution) is assumed to capture/approximate the contribution of $U_i$ to $X_i$ (under uncertainty in the structural equations). We will clarify this point in the revised paper.
>
> **Design condition 2.** Notice that from the definition of the SCM, the posterior distribution of $U_i$ only depends on the parents, i.e $U_i | X_{pa_i}$. We want VCAUSE to mimic the SCM, thus we need VCAUSE to satisfy the dependencies in the posterior distribution. Then, the GNN that parametrizes the encoder must not have hidden layers otherwise the nodes will depend on ancestors that are not the parents due to the message passing algorithm: in the iteration $k$ of a message passing node it depends on the $k$-hop ancestors. We require $k=1$ in the encoder, which refers to a GNN without hidden layers.
>
>
> Thank you for your feedback. We strongly hope that our answers have satisfied your need for clarification and alleviated your doubts.

---

### Official Review · Reviewer_ENhm · 2021-07-16

**Rating:** 6
**Confidence:** 4

**Summary:**

The paper leverages the power of variational graph autoencoders and graph neural networks and proposes a novel framework, namely Variational Causal Autoencoder, to answer the interventional query at a population level and counterfactual query at an individual level. The paper further studies the counterfactual fairness as a use case under the development.

**Limitations And Societal Impact:**

See main reviews for limiations. There is no societal impact.

**Main Review:**

In general, the paper is well written and read naturally. There are two key assumptions in the framework: 1) there are no hidden confounders in the causal graph; 2) the causal graph is available. As also said by the author at the end, both are a bit strong. Despite the two assumptions, the model first approximates the posterior of an exogenous vector Z in d dimensional for an endogenous sample X in d dimensional as well, using a graph neural network. After interventional changes or counterfactual changes on Z, the modified Z can be decoded back to the endogenous variable space using a GNN to obtain its interventional appearance or counterfactual appearance. The experiments are conducted in synthetic/semi-synthetic datasets with d = 7 at most.

-Originality
The framework that combines GNN and VGAE is novel but not surprising due to its strong assumptions. It is natural to model a graph-like data structure using GNN. The VGAE, on the other hand, is well studied in terms of manipulation of its latent space.

-Quality
The quality of the paper is good overall. It is well written and easy to follow. However, the experiments conducted are mainly from synthetic/semi-synthetic datasets. And the dimenson of data sets is relatively low (7 at most).

-Clarity
The paper is good in terms of clarity on presenting the ideas, formulating the question, and conducting experiments.

-Significance
As said above, the method is novel but carrying two strong assumptions, which is a limitation. Thus, its significance is fair to a minor problem.

-Concerns
My main concerns are:
1) The no-hidden-confounders assumption is quite strong.
2) The data sets are synthetic/semi-synthetic. The dimension is relatively low. It seems that only continuous variables are considered, no discrete variables.
3) It might be interesting to see the study on the dimension of Z_i (i.e. the parameter m). Does that have an effect on overall results?
4) The decoder might induce bias on the reconstructed sample.

**Time Spent Reviewing:**

10

---

> ### Author Response · Authors · 2021-08-10
> **Reply to Reviewer ENhm**
>
> **Strong assumptions.**  We agree with the reviewer that causal sufficiency is a strong, but also broadly adopted, assumption for causal reasoning. Access to the graph, together with the assumption of no hidden confounders,  is in general  the minimal necessary set of assumptions for causal inference from observational data [1]. Note, the graph can be obtained from domain knowledge of one of the numerous causal discovery approaches (e.g. cited in the paper in Line 52). Yet, we would like to remark that, as shown in Section 4.4., we can (mildly) relax such an assumption by grouping together those variables affected by hidden confounders (or for which the causal link is unknown) in a single node of the GNNs. This allows us to deal with a large variety of graphs in practice (see our experiments with German Credit dataset in Section 6).
> [1] Jonas Peters, Dominik Janzing, and Bernhard Schölkopf. Elements of causal inference: foundations and learning algorithms. MIT press, 2017.
>
>
> **Datasets.**  Section 5 includes experiments conducted on more diverse causal models (in terms of number of nodes and complexity of SEMs) compared to the experiments in baseline approaches. Moreover, Section 6 uses German Credit dataset, which contains continuous and discrete features, as well as confounding variables (that we group in a single node in the GNN, as we assume the absence of hidden confounders among the nodes in our causal graph). We agree, as acknowledged in the Conclusions, that further experiments assuming larger graph sizes and applications domains (e.g. biology) would be of great interest. However, we consider this to be out of the scope of the present paper and leave this as an interesting venue for future work. Yet, we believe that the contributions of the paper are significant, and will open new venues for research at the intersection of GNNs and causal inference in diverse application domains.
>
>
>
> **Dimension of Z.**  We have repeated the experiments on the triangle dataset with the non-linear and non-additive SCMs varying the dimension of the latent space of VCAUSE in the set $\{2, 4, 8, 16\}$. As we can observe in the table below, the dimension of $Z_i$ does not have a clear impact on the results for small sizes. However, as we dim($Z_i$) becomes very  large we observe indications of overfitting, especially in the extreme case dim(z)=32. We will include this table and the corresponding discussion in the Appendix for completeness.
>
> Mean and standard deviation (within brackets) for the MMD Obs., MMD Inter. and MSE CF for the-linear triangle dataset.
>
>  |dim(z)         |  MMD Obs.     |      MMD Inter.     |     MSE CF      |
>  | ------------- | ------------- |   ------------- |           ------------- |
> |2.0     |   0.047  (0.033)  |    0.052  (0.019)  |  0.103  (0.006)|
> |4.0      |  0.032  (0.023)    |  0.045  (0.016)  |  0.102  (0.006)|
> |8.0     |   0.031  (0.021)  |    0.043  (0.010)  |  0.103  (0.006)|
> |16.0   |   0.027  (0.015)    |  0.055  (0.019)  |  0.114  (0.009)|
> |32.0    |  0.026  (0.017)   |   0.093  (0.032)  |  0.124  (0.011)|
>
>
>
>
> Mean and standard deviation (within brackets) for the MMD Obs., MMD Inter. and MSE CF for the non-additive triangle dataset.
>
>  |dim(z)         |  MMD Obs.     |      MMD Inter.     |     MSE CF      |
>  | ------------- | ------------- |   ------------- |           ------------- |
> |2.0       | 0.030  (0.013) |     0.163  (0.018)   | 0.230  (0.007)|
> |4.0      |  0.032  (0.011)  |    0.170  (0.027)    |0.229  (0.006)|
> |8.0     |   0.039  (0.011)   |   0.204  (0.026)    |0.239  (0.011)|
> |16.0  |    0.052  (0.021)    |  0.219  (0.045)   | 0.242  (0.009)|
> |32.0 |     0.110  (0.131)     | 0.325  (0.074)  |  0.263  (0.014)|
>
>
>
> **Bias of the decoder.**  Would you be able to kindly clarify what you mean by “ bias on the reconstructed sample. ”? VCAUSE  indeed has "observational noise" that is not present in the true structural causal model, where the structural equations (SEs) deterministically  determine an observed variable from its exogenous variables and parents. Since VCAUSE does not have access to the true SEs, one interpret the "noise" of the likelihood  p(X|Z) as an estimate of the uncertainty in the estimated observational distribution (due to the uncertainty in the true SEs). In our synthetic experiments, we assume the likelihood X|Z is a Gaussian with mean parametrized with a NN and a  variance that we treat as a hyperparameter that we cross-validate using a grid of several (and small, as the data is assumed to be standardized) values. This is similar to the approach adopted by Karimi et al. (2020). Still, we show in Table 2 and Figure 4 (orange line) that VCAUSE predictions are close to the true value and the error is consistently small (the variance on the error is small).
>
> Thank you for your feedback. We strongly hope that our answers have satisfied your need for clarification and alleviated your doubts.

---

### Official Review · Reviewer_bpui · 2021-07-17

**Rating:** 7
**Confidence:** 4

**Summary:**

This work introduces the variational causal autoencoder (VCAUSE), a model for estimating interventional and counterfactual distributions from observational data and a known causal graph. VCAUSE builds on graph neural networks (GNNs) to propagate information across the causal graph, enabling simulation of interventions and abduction of exogenous noise for counterfactual inference.

**Limitations And Societal Impact:**

- The paper is missing a discussion of the implications of causal modelling with deep learning. How can we tell that the model is right? What can happen when people misuse the model, i.e. apply it outside the built-in assumptions? Who may be harmed by such misuse, and in what scenarios?
- The multiple claims of 'accurate' counterfactual inference may mislead practitioners to blindly trust the counterfactual predictions of this model, without knowing that this assumes the neural networks faithfully match the functional form (not just the outputs!) of the true causal mechanisms, and that the accuracy cannot be objectively measured in real-world settings. Any work dealing with counterfactuals must clearly acknowledge such limitations to reduce risks of misuse and over-interpretation, especially in high-stakes applications.

**Main Review:**

### Originality
- From my limited knowledge of the graph deep learning and causal discovery literatures (where I know GNNs have been used), this application of a GNN-based VAE for modelling a causal graphical model seems novel and inspiring.
- On the other hand, it is unclear what limitations of the existing methods this work is aiming to address. Specifically, readers are missing a compelling description of the advantages of the GNN-based VCAUSE over CVAE-based (e.g. Karimi et al., 2020) or flow-based (Khemakhem et al., 2020) approaches. Is it more general, more scalable, easier to define/implement, more stable to train...?
- Other comments:
  - CAREFL (Khemakhem et al., 2021) seems somewhat mischaracterised in the related work section: it is not evident in the text that CAREFL is not limited to two variables, and that it can model non-linear mechanisms between endogenous/observed variables.
  - It would be relevant to relate your work to Pawlowski et al. (2020). Their deep SCM framework is a generalisation of Karimi et al. (2020)'s MultiCVAE—supporting abduction-action-prediction for high-dimensional variables (e.g. images) with VAE-based, flow-based, or GAN-based causal mechanisms.
    > Pawlowski, N., Castro, D. C., Glocker, B. Deep Structural Causal Models for Tractable Counterfactual Inference. In _NeurIPS_ 2020.

### Quality
- The theoretical portion of this work is interesting, principled, and well developed. I appreciated the care in formulating and proving the model design requirements to match the behaviour of a given causal graph.
- However, the empirical benchmarking (Section 5) was limited to small, synthetic tabular datasets. While I understand the value of toy experiments in canonical conditions to validate basic model capabilities, the presented results did little to convince the reader to favour the proposed method over the baselines in realistic settings (e.g. larger, more complex graphs; low- and high-dimensional variables; noisy real-world data).
- The fairness experiment (Section 6) is not very informative due to the choice of baselines. In absolute terms, the VCAUSE-based fair classifier seems to perform well fairness-wise on this German Credit dataset. However, it has more expressive power (computational layers) than the shallow baselines, so it does not look like a fair comparison. Furthermore, no other counterfactual inference methods (e.g. MultiCVAE) were evaluated for fairness assessment, which makes the conclusions from this experiment unclear.
- The authors claim VCAUSE can 'accurately' estimate counterfactual distributions of unknown SCMs. Yet, this is impossible by definition! Counterfactuals are unidentifiable without knowledge of the structural equations because they cannot be falsified based on factual data alone—observational or even experimental (see e.g. Pearl, _Causality_, 2009, Ch. 7). Such evaluation is possible only in synthetic settings with access to a simulator (i.e. the true data-generating process). These claims are misleading and need rephrasing (see below).
- Estimating interventions in VCAUSE (Section 4.2) seems to incur a reconstruction error on the intervened variables (i.e. $do(\mathbf{X}_\mathcal{I} := \boldsymbol{\alpha})$ results in $\mathbf{X}_\mathcal{I}\neq\boldsymbol{\alpha}$), though the implications are never discussed in the paper.

### Clarity
- The paper is overall very well written and well structured. The authors present thorough additional theoretical and experimental details in the supplement, along with sharing the code.
- No justification is presented for the use of PIWAE for training. Why is it necessary? Does the model fail to converge with the vanilla ELBO?
- Minor suggestions:
  - The distinction between $\mathbf{F}=\\{f_i\\}$ and $\tilde{\mathbf{F}}=\\{\tilde{f_i}\\}$ is confusing and could be made a lot more explicit in the text.
  - [Eqs. (2–4)] As written, the integrals are computing expectations, not the intended pushforward distributions.
  - [Eq. (6) & L237] These expectations are 'double-counting' the $\mathbf{Z}$ variables. Either make them double integrals with $\iint\cdots d\mathbf{Z}\\,d\mathbf{Z}^\mathcal{I}$ or explicitly specify the indices of $\mathbf{Z}$ over which $p$ and $q_\phi$ are being evaluated.
  - The sudden switch from using $\mathbf{U}$ to $\mathbf{Z}$ in Section 4 was jarring. Consider explaining early the differences between these two sets of variables.
  - It would be helpful to include visualisations of the distributions entailed by (at least some of) the SCMs used in the experiments to give a sense of the tasks' difficulty and encoded assumptions.

### Significance
- The work presents interesting ideas that may be further extended and applied to varied types of data.
- Nevertheless, there does not appear to be strong enough discussion or evidence to convince readers to adopt this methodology rather than existing ones.
----
### Update after discussion
Under the assumption that the authors will include the discussed clarifications in the final version of the paper, I am increasing my recommendation by 1.

**Time Spent Reviewing:**

12

---

> ### Author Response · Authors · 2021-08-10
> **Reply to Reviewer bpui**
>
> **VCAUSE motivation.** The main motivation for our proposed methods are the limitations of previous work, which either i) require to train each node independently in topological order with the danger of propagating the error in cascade; and/or ii) cannot distinguish between intervening and conditioning on a zero-value. In contrast, VCAUSE encodes the causal graph information (inductive bias) into GNNs to i) distinguish the different paths in the causal graph enabling approximating the observational, interventional and counterfactual distribution at once, and ii) to enable, via the GNN message passing,  interventions (do-operator), which effectively remove the causal effect of the former parents of a variable when it is intervened upon (instead of conditioning on a zero-value). This is only possible by exploiting the (causal graph) inductive bias of the encoding and decoding GNNs of VCAUSE.
>
> **CAREFL mischaracterization.** We additionally apologize if CAREFL (Khemakhem et al., 2021) was mischaracterized in our description (we assume the reviewer refers to lines 62-63). We will rephrase such lines to make it more clear that the authors of CAREFL focus the experiments on causal inference on the  bi-variate scenarios, but the model is applicable to more complex scenarios. Also, we state in line 62 that CAREFL can model affine transformation, which are defined as:
> $x_j = exp(x_{<\pi(j)}) z_j + t_j(x_{<\pi(j)})$
>
> **Baseline.** We thank the reviewer for the suggestion Pawlowski et al. (2020)  as baseline. As to our assessment, the model seems similar to Karimi et al. (2020) while using instead normalizing flows for each node. We will happily  include this baseline in the revised version of the paper. Yet, we hope the reviewer agrees with us in the fact that such a comparison do not significantly affect the main contributions of our paper.
>
>
> **Benchmark.**  We agree that an ablation study on the scalability of current methods for causal inference (including but not limited to VCAUSE) is missing and it could definitely provide further insights about their limitations. Section 5 includes experiments conducted using more scenarios (also with higher complexity) than the baselines. For example, CAREFL only evaluates interventions/counterfactuals in root nodes (which is actually conditioning) and only in the bi-variate (multidimensional) case. Additionally, we also evaluate counterfactual fairness with the German Credit dataset, which contains heterogeneous features (e.g., discrete and continuous) as well as confounding variables (which we model as a single node). We agree, as acknowledged in the Conclusions, that further experiments assuming larger graph sizes and applications domains would be of great interest. However, we consider this to be out of the scope of the present paper and leave this, as well as an ablation study of the scalability of GNNs to larger causal graphs, as an interesting venue for future work. Still, we believe that the contributions of the paper are significant, and will open new venues for research at the intersection of GNNs and causal reasoning.
>
> **Counterfactual fairness use-case.** We would like to remark that the purpose of the analysis of Section 6 is to show that VCAUSE provides a ranking of the different approaches for fair classification which is consistent with the original counterfactual fairness paper [1]. In the case of the comparison of classifiers, we would like to stress that we consider classifiers from the same family (e.g. logistic regression) but train with different input features (which in the case of VCAUSE are the latent variables). Thus, we fail to understand why the reviewer sees this as an unfair comparison. Please classify if we are missing anything.
>
> We did not include the baselines in this analysis, as i) Section 5 shows that VCAUSE outperforms competing methods in several datasets; and ii) the available code for the baselines do not account for settings in which a node in the graph include several observed variables (which is the case of the German credit). Such a generalization is straightforward for the MultiCVAE, and thus if the reviewer considers it important, we are happy to include it in the revised version of the paper. Such an extension is unfortunately not as straightforward in the case of CAREFL and thus we consider it out of the scope of the paper.
>
> [1] Kusner, M. J., Loftus, J., Russell, C., & Silva, R. (2017). Counterfactual Fairness. Advances in Neural Information Processing Systems, 30.
>
> **Counterfactual estimation.**  We agree with the reviewer, and as shown in our experiments, that the quality of counterfactuals can only be quantified in the synthetic settings, where we have access to the true SEM. We apologize for the potentially misleading wording and will update the text to avoid potential misunderstanding from future (non-causality expert) readers of the paper, and thus the misuse of our (and any other) approach for counterfactual reasoning.
>
>
> **Accurate counterfactual inference can be misleading.** We will strongly put emphasis on the fact  that counterfactual inference can only be evaluated in synthetic scenarios in which we have access to the true SEM.
>
> **Reconstruction error.**  VCAUSE  indeed has "observational noise" that is not present in the true structural causal model, where the structural equations (SEs) deterministically determine an observed variable from its exogenous variables and parents. Since VCAUSE does not have access to the true SEs, one interpret the "noise" of the likelihood  p(X|Z) as an estimate of the uncertainty in the estimated observational distribution (due to the uncertainty in the true SEs). In our synthetic experiments, we assume the likelihood X|Z is a Gaussian with mean parametrized with a NN and a variance that we treat as a hyperparameter that we set to a small value, as the data is assumed to be standardized. This is similar to the approach adopted by Karimi et al. (2020). Still, we show in Table 2 and Figure 4 (orange line) that VCAUSE predictions are close to the true value and the error is consistently small (the variance on the error is small).
>
> **Objective function.** The PIWAE was chosen because it has been shown to work generally better than the standard ELBO (when interested in jointly solving both density estimation and posterior inference tasks), as expected since the latter is a lower bound on the objective function. We will add this information in the camera-ready version. Additionally, in the GitHub repository we provide the code to train VCAUSE using the ELBO, IWAE or PIWAE, in case there is interest from future practitioners. Regarding the number of iterations, we assumed a large enough number of epochs for training loss to converge, and implemented  early stopping to prevent overfitting the training data. This was our practice not only for VCAUSE but also for the baselines.
>
> **Equations 2-4.** We believe there is a misunderstanding. Eqs. (2-4) simply marginalize out the exogeneous variables U, thus the resulting distributions are over the corresponding endogenous variable X.
>
> **Typos.** We noted the typos and will upload them in the final version
>
> **U versus Z.** Let us try to clarify the difference between the true exogenous variables $U$ and the latent variable of VCAUSE $Z$. We remark that  the latent variables $Z$ play a similar role to the exogenous variables $U$, however we emphasize that $Z$ does not need to correspond to the exogenous variables. As the structural equations and the true distribution of the exogenous variables $U$ is unknown, there is no way we can ensure that  VCAUSE latent variables $Z$ correspond to the true exogenous variables. That said, what we assume is that there is one independent  latent variable $Z_i$ for every observed variable $X_i$ capturing all the information of $X_i$ that cannot be explained by its parents. Thus, since $X_i$  is in turn a (deterministic) function of its parent and its exogenous variable $U_i$, the encoded $Z_i$ (actually its posterior distribution) is assumed to capture/approximate the contribution of $U_i$ to $X_i$ (under uncertainty in the structural equations). We will further clarify this point in the revised paper.
>
> **Visualization of distributions of datasets.** We agree with the reviewer and will include the figures with the distribution of the endogenous variables of the different datasets in the reviewed version of the paper.
>
> **Implications of causal modeling.** Thank you very much for this important point raised. We certainly agree the potential social impact of counterfactual generation for decision making is of great importance. As noted in the conclusion, we suggest evaluating the sensitivity of VCAUSE to the violation of assumptions as hidden confounders, in order to avoid its misuse. We are happy to elaborate more on the general risks of counterfactual generation.
>
>
> Thank you for your feedback. We strongly hope that our answers have satisfied your need for clarification and alleviated your doubts.

---

> > ### Comment · Reviewer_bpui · 2021-08-25
> > **Follow-up**
> >
> > I thank the authors for the response. I am willing to increase my score by 1 assuming the authors will integrate the referred clarifications in the final version.
> >
> > Below I follow up on a few outstanding points:
> >
> > **VCAUSE motivation.**
> > i) I disagree; my understanding is that every node in both CAREFL and MultiCVAE is trained with real data as inputs and outputs, without propagating errors across the graph.
> > ii) Intervening on such models is also trivial (just replace the relevant node). I don't see where the mentioned zero-conditioning comes from.
> >
> > **CAREFL mischaracterisation.**
> > That transformation is affine in the noise $z_j$ but non-linear in the parents $x_{<\pi(j)}$, which is what matters regarding expressiveness of the SCM.
> >
> > **Baseline.**
> > I meant it could be discussed as related work; I agree there's no need for an additional experimental comparison.
> >
> > **Benchmark.**
> > While I believe the idea has merit, I maintain that the empirical evidence of its benefit over the alternatives in practical settings is very limited.
> >
> > **Counterfactual fairness use-case.**
> > I understand and agree with the first point. Regarding input features, we have e.g. $h_\mathrm{full}(\mathbf{X}) = \mathrm{LR}(\mathbf{X})$, while $h_\mathrm{VCAUSE}(\mathbf{X}) = \mathrm{LR}(\mathrm{VCAUSE}(\mathbf{X}))$, where $\mathrm{VCAUSE}(\cdot)$ is a multi-layer neural network.
> > My comment suggested that a fairer comparison would allow a comparable amount of non-linear feature extraction for the baselines.
> >
> > **Reconstruction error.**
> > The question was: why not just overwrite the intervened variables $\mathbf{X}_\mathcal{I}$ with $\boldsymbol{\alpha}$ instead of keeping the imperfect reconstructions $\hat{\mathbf{X}}_\mathcal{I}$?
> >
> > **Equations 2–4.**
> > My point is that
> > $$\int \mathbf{F(U)} p(\mathbf{U}) d\mathbf{U} = \mathbb{E}[\mathbf{F(U)}] = \mathbb{E}[\mathbf{X}] \neq p(\mathbf{X}),$$
> > and analogously for the others.
> > It is messy to deal with densities in such contexts—you could replace $\mathbf{F(U)}$ by $\delta_{\mathbf{F(U)}}(\mathbf{X}) / \left|\frac{d\mathbf{F(U)}}{d\mathbf{X}}\right|$, which would be more correct but less readable.
> > Suggestion: What you mean is that $P(\mathbf{X})$ is the [pushforward](https://en.wikipedia.org/wiki/Pushforward_measure) of $P(\mathbf{U})$ through $\mathbf{F}$, which could be written more concisely as $P(\mathbf{X}) = \mathbf{F}_\sharp[P(\mathbf{U})]$, for example.

---

> > > ### Author Response · Authors · 2021-08-30
> > > **Reply to Reviewer bpui**
> > >
> > > Thank you for your reply.
> > >
> > >
> > > **VCAUSE motivation.** First, we would like to point out our arguments i) concerned MultiVAE and ii) concerned CAREFL:
> > >
> > > i) Indeed MultiVAE trains each node with real data independently. This precisely can be problematic whenever we are interested in evaluating the joint distribution, i.e. want to jointly generate observational (or interventional/counterfactual) samples from all (or a subset of) nodes. For each node $X_i$, the mismatch between the true and generated distribution can cause errors that propagate to the distribution of its descendents;
> > >
> > > ii) With regards to CAREFL, flow architectures like neuronal spline flows (NSF) allow to capture arbitrarily complex graphs using a topological ordering of the nodes in a DAG. By using these flow architecture in CAREFL, the variable $Z_i$ of a node $X_i$ depends on all its previous nodes---previous in terms of causal (i.e.. topological) order--- as detailed in Section 3.1 Implementation (1.) in [1]. When intervening on $X_i$ the model does not break the dependency of $Z_i$ on the causal ancestors, i.e. the interventional distribution of the descendants of $X_i$ does depend on the value of v (see Appendix E.1 [2]). We would additionally like to remark that CAREFL--  built on autoregressive flows --  uses causal knowledge in the form of causal (i.e.. topological) ordering of nodes but not the causal graph structure. While causal knowledge in a graph is modeled by the absence of edges (see [3]), CAREFL is unable to exploit this knowledge fully as it reduces a causal graph to its causal ordering (which may not be unique).
> > >
> > >
> > >
> > > [1] Durkan, Conor, et al. "Neural spline flows." Advances in Neural Information Processing Systems 32 (2019): 7511-7522.
> > >
> > > [2] Khemakhem, Ilyes, et al. "Causal autoregressive flows." International Conference on Artificial Intelligence and Statistics. PMLR, 2021.
> > >
> > > [3] Pearl, Judea. "Causal inference in statistics: An overview." Statistics surveys 3 (2009): 96-146.
> > >
> > > **Benchmark**. Thank you again for the suggestions for further experiments. As stated earlier, we acknowledge that larger experiments can be very insightful. At the same time the diversity and complexity of SCMs we present in our empirical evaluation is not only vast, but also exceeding the work of the baselines we compare to. As stated in our previous comment, we believe that the empirical evaluation addresses different problems of practical concern (e.g. non-linear and non-additive structural equations, heterogeneous data, heterogeneous nodes, real world data)
> > >
> > > **Counterfactual fairness use-case.** Thank you for your clarification, we believe we understand your point of view. Do you mean to suggest e.g. to use a normal VAE trained on the full, unaware and fair set of inputs and use the latent Z of the respective VAE as input to LR and SVM?
> > > In this case, the accuracy of the LR/SVM using the latent variable Z induced by the different VAE may or may not be higher than the corresponding LR/SVM without feature extraction (as reported in the paper). This is hard to tell since the VAE is not trained for the classification task. We can certainly include this in the revised version of the paper for completeness, if you deem necessary. At the same time we would like to emphasize that VCAUSE was not primarily developed for counterfactual fair classification. Section 6 of the paper serves to show an additional application of our method.
> > >
> > > **Reconstruction error.** Thank you for the clarification. As detailed in Appendix D3, we measure reconstruction error (MeanE, StdE, MSE, SSE) on descendants des($X_{\mathcal{I}}$) of intervened-on variables only. With that the reconstruction of the intervened-on variable does not distort the evaluated metrics.
> > >
> > > **Equations 2–4**. Thank you for the suggestion to improve the notation. We agree and will update our notation as you suggested, while we will assure that definitions remain as readable as possible.
> > >
> > > We hope that our answers have clarified your concerns.

---

### Official Review · Reviewer_7HjM · 2021-07-17

**Rating:** 4
**Confidence:** 4

**Summary:**

The authors propose a framework for the use of variational graph autoencoders for causal inference in settings without hidden confounders when observational data and the causal graph are available. The framework aims to mimic the properties of the matching Structural Causal Model (SCM), including the ability to evaluate observational, interventional and counterfactual queries. To evaluate their proposed framework, the authors perform an experimental evaluation in estimating the outcomes of causal queries on multiple small synthetic causal graphs and in assessing counterfactual fairness.

**Ethical Concerns:**

I have no concerns.

**Limitations And Societal Impact:**

**Societal Impact**:
- I see no immediate reason for concern.

**Questions/suggestions for the authors**:
- In Section 2.1, the authors state that "Every SCM M is associated with a directed acyclic graph (DAG) [...]." This is, to the best of my knowledge, not true - SCMs are not by definition acyclic and can in the general case include cycles [6]. Is VCAUSE only applicable to acyclic settings? If yes, please add the acyclicity assumption to the manuscript.

- Please add the missing compute time details and a comparison of computational considerations between VCAUSE and existing methods.

- Introduction, 3rd paragraph: the sentence "[...] but also on how these relationships are (i.e., on the structural equations)." is missing a verb, e.g. "how these relationships are mediated (e.g., in the structural equations)".

- In Table 2 (main) and Table 12 (appendix), the caption indicates that all metrics are given in percentages. Please provide absolute values instead (or at least the reference values used to compute the percentages).

- In Table 2 (main) and Table 12 (appendix), some values are shown in bold print. However, it is not stated what the bold print indicates. Does the bold print indicate statistical significance? If so, compared to what reference?

- Given that the authors claim "(...) *significantly* more robust results than competing methods" in the presented conclusion, the statistical significance of all results should be reported in the manuscript (p-values) - in particular for Table 2.

- Line 325 to 327: "Note that to perform interventions, CAREFL sets the parents of the intervened variables to zero, which may not completely severe the causal paths to the intervened nodes. " - I assume CAREFL sets the *contribution of* the parent variables (and not the parent variables themselves) to the intervened node to zero (which would correspond to Pearl's do-operator). Can the authors explain why this would not sever all causal paths to the intervened nodes? If all incoming contributions are zeroed, to my understanding no other causal paths to the intervened node can exist.

- Line 327: "which may not completely severe [should be *sever*] the causal paths to the intervened nodes."

[1] Ke, Nan Rosemary, et al. "Learning neural causal models from unknown interventions." arXiv preprint arXiv:1910.01075 (2019).

[2] Peters, Jonas, et al. "Causal discovery with continuous additive noise models." (2014).

[3] Khemakhem, Ilyes, et al. "Causal autoregressive flows." International Conference on Artificial Intelligence and Statistics. PMLR, 2021.

[4] Monti, Ricardo Pio, Kun Zhang, and Aapo Hyvärinen. "Causal discovery with general non-linear relationships using non-linear ica." Uncertainty in Artificial Intelligence. PMLR, 2020.

[5] Stone, Richard. "The assumptions on which causal inferences rest." Journal of the Royal Statistical Society: Series B (Methodological) 55.2 (1993): 455-466.

[6] Bongers, Stephan, et al. "Foundations of structural causal models with cycles and latent variables." arXiv preprint arXiv:1611.06221 (2016).

**Main Review:**

**Originality**:
- The question of how to accurately answer causal queries with machine learning models from observational data given knowledge of the underlying causal graph is of practical relevance in some use cases (see limitations outlined in "Significance" below), and not yet extensively studied in literature.
- The application of graph variational autoencoders to this setting is to the best of my knowledge novel. However, the manuscript's exposition is relatively sparse in details as to why this direction of research would be expected to lead to performance gains.

**Quality**:
- Given that VCAUSE is proposed as a broadly-applicable tool for answering causal queries from observational data when the underlying causal graph is available, I am concerned that the presented evaluation is not representative enough to substantiate claims that VCAUSE is in general better able to estimate observational, interventional and counterfactual distributions. In particular, the experimental evaluation only evaluated graphs with 3 and 7 nodes, whereas in practice significantly larger graph sizes are commonly encountered (e.g., in biology where structures with 100s to 1000s of nodes are not uncommon). I believe an evaluation on a larger set of diverse settings with varying graph properties (number of nodes, connection sparsity, ...) would be necessary (i) to substantiate broad claims in the general case and (ii) to better understand the relative performance of VCAUSE under varying assumed graphical structures.

- A better understanding of the importance of the properties of the modelled SCM on VCAUSE would be particularly important because Table 2 indicates that the relative performance of VCAUSE appears to depend strongly on the underlying SCM.

- Several relevant reference baselines are missing from the experimental comparison (Table 2), e.g. learning the parameters of an SCM directly using neural networks [1], additive noise models [2] which can be parameterized by different regressor classes (see e.g. [3]) and non-linear structural equation models (SEMs) [4].

- Different approaches were used for hyperparameter optimization for each of the compared baselines (Appendix D.2). It is therefore unclear if differences in observed performance (Tables 2 and 12) are due to inherent properties of the models or the different hyperparameter optimization strategies and budgets. The same hyperparameter optimization strategy with a fixed hyperparameter optimization budget should have been employed to enable a fair comparison of methods on equal grounds.

- In addition, there are several VCAUSE hyperparameters for which only the final selected values are presented (Appendix D.2) but no details are given as to *how* they were selected (learning rate, epochs, K, latent variable dimension, number of neurons, ...). Presumably, these values are the result of manual optimization - if so, this should be transparently stated in the manuscript including information as to whether or not held-out performance results were used to optimize these hyperparameters (risk of overfitting).

- The application of VCAUSE to use cases in counterfactual fairness (Section 6) is interesting but no state-of-the-art baselines are presented for comparison. It is therefore unclear if VCAUSE performs better, equally well or worse in the counterfactual fairness setting than other methods that parameterise an SCM. In addition, uncertainty and statistical significance metrics for the experiments in Section 6 are missing.

**Clarity**:

- In terms of clarity, my main consideration is that the motivational element appears to be missing in the paper's story line. Why do we expect that mimicking an SCM using a graph VAE is a more efficacious approach than alternatives (and therefore embark on this line of research)?

- In Table 2 (main) and Table 12 (appendix), the caption indicates that all metrics are given in percentages. However, the authors do not state what reference values were used to compute the percentages. In addition, even if the reference values were indicated, I would generally prefer to be shown the absolute values, since some metrics are not on a linear scale (e.g. MSE) which makes interpretation as percentages difficult.

- The computational characteristics of VCAUSE and how they compare to existing alternatives are missing. Compute time details are in general missing from the manuscript (the Paper Checklist refers to Appendix D but no compute time information is provided therein).

**Significance**:
- The overall impact of the presented work is limited because of the restrictive assumptions (availability of a faithful causal graph and no hidden confounding) that hold true in a very limited set of settings and can in practice not be verified from data alone [5]. For practical use cases that VCAUSE would be considered for, there therefore exists the risk that the assumptions would likely be at least partially violated.

- In addition, the manuscript's claims are only substantiated with experimental evidence for small directed acyclic graphs (DAGs), and it is therefore currently unclear if VCAUSE offers relative benefits in settings with larger numbers of nodes.

- Given the absence of several baselines in the experimental comparison, I am unsure if the complexity of processing the causal graph through a graph neural network (GNN) is necessary. What motivates this approach over simpler existing ones (e.g. parameterising the SCM using neural networks [1] or additive noise models [2]) that would justify the added complexity? What is the inductive bias introduced by VCAUSE that one would expect to lead to performance benefits?


**Time Spent Reviewing:**

4

---

> ### Author Response · Authors · 2021-08-10
> **Reply to Reviewer 7HjM (1)**
>
> **Motivation for using VCAUSE.** The main limitation of previous work is either that i) they require to train each node independently in topological order with the danger of propagating the error in cascade (MultiVAE); and/or ii) they cannot distinguish between intervening and conditioning on a zero-value (CAREFL). In contrast, VCAUSE encodes the causal graph information (inductive bias) into GNNs to i) distinguish the different paths in the causal graph, which allows to approximate the observational, interventional and counterfactual distribution at once, and ii) to enable, via the GNN message passing,  interventions (do-operator), which effectively remove the causal effect of the former parents of a variable when it is intervened upon (instead of conditioning on a zero-value). This is only possible by exploiting the (causal graph) inductive bias of the encoding and decoding GNNs of VCAUSE.
>
> **DAG Assumption.** We are aware that, as pointed out by the reviewer, not all causal graphs are DAGs. At the same time, this is a common assumption (present in the baselines [1, 2] as well as in the literature referenced by the reviewers) that we also adopt. We will make sure to clarify this point in Section 2.1. We would like to stress that, as show in Section 4.4., we can relax such an assumption to handle undirected/unknown causal edges by grouping together those variables affected by hidden confounders (or for which the causal link is unknown) in a single node of the GNNs. This allows us to (mildly) relax our causal sufficiency assumption and deal with a large variety of graphs in practice (see our experiments with German Credit dataset in Section 6). Extensions to other causal graphs, e.g., including bi-directed edges and other cycles, are definitely interesting venues for future work.
>
> [1] Karimi, A. H., Von Kügelgen, J., Schölkopf, B., & Valera, I. (2020). Algorithmic recourse under imperfect causal knowledge: a probabilistic approach. Advances in Neural Information Processing Systems, 265--277.
>
> [2] Khemakhem, I., Monti, R., Leech, R., & Hyvarinen, A. (2021, March). Causal autoregressive flows. In International Conference on Artificial Intelligence and Statistics (pp. 3520-3528). PMLR.
>
> **Larger causal graphs.** We agree that an ablation study on the scalability of current methods for causal inference (including but not limited to VCAUSE) is missing and it could definitely provide further insights about their limitations. Section 5 includes experiments conducted on more diverse causal models (in terms of number of nodes and complexity of SEMs) compared to the experiments in baseline approaches. Moreover, Section 6 uses German Credit dataset, which contains continuous and discrete features, as well as confounding variables (that we group in a single node in the GNN, as we assume the absence of hidden confounders among the nodes in our causal graph). We agree, as acknowledged in the Conclusions, that further experiments assuming larger graph sizes and applications domains (like the one pointed out by the reviewer) would be of great interest. However, we consider this to be out of the scope of the present paper and leave this, as well as an ablation study of the scalability of GNNs to larger causal graphs, as an interesting venue for future work. Yet, we believe that the contributions of the paper are significant, and will open new venues for research at the intersection of GNNs and causal inference in diverse application domains like biology.
>
> **VCAUSE performance for different SCMs.** We agree with the reviewer that it would be interesting to further discuss how increasing the complexity in SCMs (e.g., by assuming non-linear SEs) affects the performance of VCAUSE and also the baselines. The deterioration of VCAUSE performance as we increase the complexity of the structural equations is due to the limited expressiveness of the GNN message computation functions (especially the aggregation function), which has already been discussed (and partially addressed) in the GNN literature [1]. VCAUSE could benefit from these advances to handle complex SCMs. We will include this (extended) discussion in a new section addressing the practical limitations of VCAUSE in the revised version of the paper.
>
> [1]  Corso, Gabriele, et al. "Principal neighbourhood aggregation for graph nets." arXiv preprint arXiv:2004.05718 (2020).
>
>
> **Missing baselines.**  We have compared our approach with two of the most recent methods that  completely fit our task: modeling interventional and counterfactual queries. We are aware there are other methods that are suited only for interventions [2] or that are focused on evaluating structure learning [3, 4] (i.e., causal discovery as opposed to causal inference). However,  to the best of our knowledge,  they do not seem fully suitable for solving the wide range of counterfactual and interventional queries in diverse SCMs (in terms of structural equations, interventions, variable types, etc.) considered in our experiments. In particular, [5] assumes all observed (endogenous) variables to be categorical; [3] consider only for real-valued observations (endogenous variables) and do not solve causal queries but instead focus on causal discovery;  and [4] focuses on bivariate causal discovery.
>
> [2] Murat Kocaoglu, Christopher Snyder, Alexandros G. Dimakis, and Sriram Vishwanath. 2018.434CausalGAN: Learning Causal Implicit Generative Models with Adversarial Training. InPro-435ceedings of the International Conference on Learning Representations (ICLR), Vol. 6.
>
> [3] Peters, Jonas, et al. "Causal discovery with continuous additive noise models." (2014).
>
> [4] Monti, Ricardo Pio, Kun Zhang, and Aapo Hyvärinen. "Causal discovery with general non-linear relationships using non-linear ica." Uncertainty in Artificial Intelligence. PMLR, 2020.
>
> [5] Ke, Nan Rosemary, et al. "Learning neural causal models from unknown interventions." arXiv preprint arXiv:1910.01075 (2019).
>
>
> **Hyperparameter tuning.**  We would like to clarify that Section 5, in which all configurations were tested using 10 independent seeds, shows the results of a fair comparison between methods.  We have cross-validated the architectures for all models considered (both VCAUSE and CAREFL) using a similar number of configurations (we used the default hyperparameters of ADAM optimizer for the three models.). In total we consider 180 NN architecture configurations for both VCAUSE and CAREFL (which range to a similar number of parameters). For MultiCVAE,  we use the best configuration of hyper-parameters reported in the original paper (as we reuse their experiment setting and code). In addition, for VCAUSE we have also tested the configurations with (10%, 20%) and without adjacency drop-out, but all the experiments
>
> [6] Karimi, Amir-Hossein, et al. "Algorithmic recourse under imperfect causal knowledge: a probabilistic approach." arXiv preprint arXiv:2006.06831 (2020).
>
> [7]  Corso, Gabriele, et al. "Principal neighbourhood aggregation for graph nets." Advances in Neural Information Processing Systems (2020).
>
>
> **VCAUSE objective.** The PIWAE was chosen because it has been shown to work generally better  than the standard ELBO (when interested in jointly solving both density estimation and posterior inference tasks). This is expected since the latter is a lower bound on the objective function. We will clarify this information in the camera-ready version. Additionally, in our GitHub repository we provide the code to train VCAUSE using the ELBO, IWAE or PIWAE, in case there is interest from future practitioners. Regarding the number of iterations, we assumed a large enough number of epochs for training loss to converge, and implemented early stopping to prevent overfitting the training data. This was our practice not only for VCAUSE but also for the baselines.
>
> **Counterfactual fairness use-case.** We would like to remark that the purpose of Section 6 is to show a practical application of the proposed algorithm. We did not include the baselines in this analysis, as i) Section 5 shows that VCAUSE outperforms competing methods in several dataset; and ii) the available code for the baselines do not account for settings in which a node in the graph includes several observed variables (which is the case of the German credit modeling). Such a generalization is straightforward for the MultiCVAE, and thus if the reviewer considers it important, we are happy to include it in the revised version of the paper. Unfortunately, such an extension is not as straightforward in the case of CAREFL and thus we consider it out of the scope of the paper. Note that the results of Section 6 provide a ranking of the different approaches for fair classification which is consistent with the original counterfactual fairness paper [8].
>
> Regarding the statistical significance of these results, we have run this experiment using a 5-fold split of the data, and in all cases the resulting ranking (as well as f1-score and fairness values) is consistent with the results in the paper. In the revised version of the paper, we plan to additionally  include a Critical Difference (CD) diagram of the classifiers rankings based on the Wilcoxon significance test [9] using 5-fold CV on German dataset x 10 random seeds for initializing  VCAUSE to further demonstrate the statistical significance of these results.
>
> [8] Kusner, M. J., Loftus, J., Russell, C., & Silva, R. (2017). Counterfactual Fairness. Advances in Neural Information Processing Systems, 30.
>
> [9] J. Demšar. Statistical comparisons of classifiers over multiple data sets. J. Mach. Learn. Res., 7:1–30, December 2006

---

> > ### Author Response · Authors · 2021-08-10
> > **Reply to Reviewer 7HjM (2)**
> >
> > **Evaluation metrics.** We would like to clarify that we do not report any relative metric that requires a reference value. All the metrics reported are absolute metrics, as can be verified in Equations (8-13) in Appendix D.3. We just multiply them by 100 for readability purposes. We will make sure to clarify this point in the revised paper.
> >
> > **Empirical result discussion.**  Bold-print in the tables indicates the results whose mean is better than one standard deviation from the mean of other methods. By significantly more robust, we mean that the variance of the errors is low, which means we are not making big errors as opposed to the baselines. We will revisit the presentation of our results to make sure that we are coherent and clear in our presentation and also that we follow NeurIPS guidelines in this regard.
> >
> > **Time Complexity.** We acknowledge that the time complexity analysis of the experiments was missing. We have re-run all the experiments to measure time complexity. Below we show the time (in minutes) it takes to train the models for different configurations. We observe that while VACUSE seems to have a larger mean (also the variance), the T-tests comparing the times (also below) indicate that there are not statistically significant differences in the mean times (at significance levels clearly below 0.05). We will include these results in the revised version of the paper.
> >
> > | Model      |    Time mean (std) in minutes |
> > | ------------- | ------------- |
> > | CAREFL   |  23.987996 (12.259490) |
> > | MultiCVAE  | 24.399543 (10.368547) |
> > | VCAUSE   |  33.196338 (18.144085) |
> >
> >
> > p-value of the T-test for VCAUSE and CAREFL: 0.0000
> >
> > p-value of the T-test for CAREFL and MultiCVAE: 0.0290
> >
> > p-value of the T-test for VCAUSE and MultiCVAE: 0.0001
> >
> >
> > **Intervention versus conditioning.** CAREFL is based on standard neural networks, for which the dimension of the input is fixed at the beginning. As a consequence, when an intervention is performed usually the intervened-on variables are set to 0, which cannot be distinguished from conditioning on 0. In contrast, VCAUSE relies on the message passing algorithm, which distinguishes between intervention (there is no message passing to the intervened-on variable) from conditioning (there is a message passed to the intervened-on variable).
> >
> > In the example in Figure 3 corresponding to the causal graph G in Figure 1, we initially have flow of information from $X_2$  ->$X_3$ and $X_1$ -> $X_2$ -> $X_3$. An intervention of the form $X_2=\alpha$ is computed in VCAUSE as follows. First, by removing in the encoder GNN the edge $X_1$ -> $X_2$ and input to the encoder $D_1 =$\{$x_1, \alpha , x_3$\}, we obtain the distribution of the latent variable corresponding to the intervened $X_2$ as  $Q$\($Z^\prime_2 |  \alpha$\) that only depends on $\alpha$.  Then, the effect of the intervention is propagated (in the decoder) to  $X_3^\prime$, which will receive messages via the paths $Z_1$ -> $X_3$, $Z_2^\prime$ -> $X3^\prime$.  This contrast with conditioning on $X_2=\alpha$, which would result in  $Q$\($Z_2^\prime |  \alpha, X_1$\) and  computation of $X_3$ which will receive messages via paths $Z_1$ -> $Z_3$, $Z_2$ ->  $Z_3$, and $Z_1$ -> (via $Z_2$) -> $Z_3$.
> >
> >
> >
> > **Typos.** We thank the reviewers for uncovering some typos. They will be corrected in the final version of the paper.
> >
> > Thank you for your feedback. We strongly hope that our answers have satisfied your need for clarification and alleviated your doubts.

---

### Decision · Program_Chairs · 2021-09-28

**Decision:**

Reject

**Comment:**

Two reviewers recommend accepting the submission and two reviewers recommend rejecting the submission. Reviewer 7HjM raises several important concerns: 1) the motivation for the work is missing from the story line, 2) the synthetic experiments show that the proposed method performs better some times and worse others, but without explanation as to why, and 3) the real data experiment does not justify the claims of the proposed method being a broadly-applicable tool. Reviewer bpui also points out limitations of the experiments, as well as many positive aspects of the work, which I agree with. However, I find the submission’s merits are not sufficient to warrant acceptance to a top-tier machine learning conference in light of aforementioned limitations.

**Consistency Experiment:**

NeurIPS has a long history of experimentation. In 2014, NeurIPS ran an experiment in which 10% of submissions were reviewed by two independent committees to quantify the randomness in the review process. This year, we repeated a variant of this experiment to see how the quality of the review process has changed over time.  This paper was part of the experiment and was therefore assigned to two committees (consisting of reviewers, an Area Chair, and a Senior Area Chair) that reached independent decisions.  If both committees made the same recommendation, this recommendation was followed. If a single committee recommended acceptance, the paper was accepted (with the exception of a few cases in which the other committee identified what we considered a fatal flaw, e.g., an error in a key result).

Both committees reached the same decision: **Reject**

The other committee assigned to the paper recommended **Reject**.  You can find the other set of reviews, along with any follow up discussion with the authors here:
https://openreview.net/forum?id=UPFWUKcdah